# Computer-Aided Detection False Positives in Colonoscopy

**DOI:** 10.3390/diagnostics11061113

**Published:** 2021-06-18

**Authors:** Yu-Hsi Hsieh, Chia-Pei Tang, Chih-Wei Tseng, Tu-Liang Lin, Felix W. Leung

**Affiliations:** 1Division of Gastroenterology, Department of Internal Medicine, Dalin Tzu Chi Hospital, Buddhist Tzu Chi Medical Foundation, Chiayi 62247, Taiwan; franktg@hotmail.com (C.-P.T.); CWtseng2@gmail.com (C.-W.T.); 2School of Medicine, Tzu Chi University, Hualien City 97004, Taiwan; 3Department of Management Information Systems, National Chiayi University, Chiayi 60054, Taiwan; tuliang@mail.ncyu.edu.tw; 4Sepulveda Ambulatory Care Center, Veterans Affairs Greater Los Angeles Healthcare System, North Hills, CA 91343, USA; Felix.Leung@va.gov; 5David Geffen School of Medicine at University of California at Los Angeles, Los Angeles, CA 90024, USA

**Keywords:** artificial intelligence, computer-aided detection, colonoscopy, false positive, water exchange

## Abstract

Randomized control trials and meta-analyses comparing colonoscopies with and without computer-aided detection (CADe) assistance showed significant increases in adenoma detection rates (ADRs) with CADe. A major limitation of CADe is its false positives (FPs), ranked 3rd in importance among 59 research questions in a modified Delphi consensus review. The definition of FPs varies. One commonly used definition defines an FP as an activation of the CADe system, irrespective of the number of frames or duration of time, not due to any polypoid or nonpolypoid lesions. Although only 0.07 to 0.2 FPs were observed per colonoscopy, video analysis studies using FPs as the primary outcome showed much higher numbers of 26 to 27 per colonoscopy. Most FPs were of short duration (91% < 0.5 s). A higher number of FPs was also associated with suboptimal bowel preparation. The appearance of FPs can lead to user fatigue. The polypectomy of FPs results in increased procedure time and added use of resources. Re-training the CADe algorithms is one way to reduce FPs but is not practical in the clinical setting during colonoscopy. Water exchange (WE) is an emerging method that the colonoscopist can use to provide salvage cleaning during insertion. We discuss the potential of WE for reducing FPs as well as the augmentation of ADRs through CADe.

## 1. Introduction

Missed lesions account for 57.8% of interval colorectal cancers (i.e., cancers that occur within 3–5 years after a negative colonoscopy) [1]. To reduce incidences of missed lesions and interval cancers, measures were proposed to improve the quality of colonoscopies. One of the most important quality metrics is the adenoma detection rate (ADR), defined as the proportion of patients with at least one adenoma [2].

Artificial intelligence (AI) is being used in the computer-aided detection (CADe) and diagnosis (CADx) of polyps [3]. Randomized controlled trials (RCTs) showed CADe-assisted colonoscopy significantly increased the ADR [4,5,6,7,8]. A meta-analysis confirmed that the ADR was significantly higher in the CADe group than in the conventional group (36.6% vs. 25.2%; RR, 1.44; 95% confidence interval, 1.27–1.62; *p* < 0.01; *I*^2^ = 42%) [9].

An accompanying limitation of the CADe is false positives (FPs), which occur when the algorithm identifies a “polyp” that the endoscopist would disagree with. FPs were ranked 3rd in importance among 59 future research questions related to CADe [10]. Therefore, we conducted this systemic review on the definitions, causes, and adverse effects of the CADe FPs. We assessed CADe-overlaid video analyses, RCTs using real-time CADe to enhance polyp detection during colonoscopies, and studies that used FPs as the primary outcome. We also reviewed water exchange (WE) colonoscopy, a novel insertion method that may help to decrease FPs. We test the hypothesis that the systematic review of the literature on FPs will yield insight into methods of managing and limiting the adverse effects of this drawback of CADe.

## 2. Method

We performed a systematic review of the literature by searching PubMed with the following string: (automatic polyp detection OR computer-aided detection OR deep learning OR artificial intelligence) AND colonoscopy in the past four years (Jan. 2017 to Apr. 2021). The search was performed for titles, abstracts, and keywords. We included full-text articles in English. The exclusion criteria were non-research reports (i.e., systematic reviews, editorials, or case reports), research not related to artificial intelligence, not focusing on colonoscopy (e.g., computed tomography colonography, capsule endoscopy, chatbot, etc.), not related to polyp detection (e.g., CADx, regulatory issues, robotic colonoscopy, quality optimization, etc.), not applying real-time video analysis, or not reporting FPs. For clinical studies, non-RCTs were excluded. We identified 9 articles on the applications of CADe based on deep learning for the real-time detection of polyps on colonoscopy videos, 6 articles on RCTs comparing colonoscopies with or without CADe to assist in polyp detection, as well as 2 CADe-overlaid video analysis studies using FPs as the primary outcome (Figure 1). We also turned to one of our most recent reviews on WE colonoscopies and tabulated 3 articles on RCTs comparing WE with air insufflation that reported FP-related procedural data [11].

## 3. Definition of False Positives

False positives have various definitions across different studies (Table 1). In general, the term refers to computer prompts indicating polyps that the endoscopist does not consider to be polyps [12,13]. Some investigators have used false alarms to define tracking boxes on non-polyp structures that were continuously tracked [7,8], while other authors ignored brief false positives [4,12]. Variations in practices have led to inconsistent reports on the frequency of FPs. The CADe-assisted detection of colon polyps has reported an average of 0.071 to 0.201 FPs per colonoscopy [4,5,6,7,8,14]. However, studies using FPs as the primary outcome reported an average of 26.3 to 27.3 FPs per colonoscopy, despite having extracted videos from the same RCT [5] or having used a similar CADe model to that of a previous RCT [7].

Defining FPs based on the duration of time is an objective way of classifying FPs. However, the threshold required for reporting FPs is unsettled. One report suggested that only FPs > 2 s be reported [13], and another only reported FPs > 1 s [15], while the majority of FPs (i.e., more than 90%) lasted <0.5 s [13]. It is unknown whether ignoring the transient FPs (i.e., those lasting for <1 or 2 s) would increase the risk of missing a real polyp. A recent report on colonoscopic video analysis with CADe showed that missed polyps had a shorter appearance time (defined as the interval on the video between appearance and disappearance of a polyp) than detected polyps had [16]. Future prospective studies that explore every flash of a prompt, including those lasting <0.5 s, would be necessary to answer the question of whether those FPs could be discarded or not. Defining FPs based on time can also be influenced by the endoscopist’s technique, such as the speed of withdrawal.

The definition of the presence of an FP is dependent on the judgment of the endoscopist. For studies analyzing CADe-overlaid videos, FPs were judged by a single expert reader or through the consensus of 2–3 reviewers [12,15,18]. Thus, there is room for errors due to subjectivity in these definitions.

Although the frequency of FPs was reported as per-colonoscopy, RCTs evaluating CADe systems did not employ the algorithm during insertion [4,6,7,8,14], and the post hoc analysis of videos for FPs invariably involved only the withdrawal phase [12]. It is conceivable that more FPs would be observed during insertion, where the bowel lumen is supposed to be kept minimally distended and the bowel contents have not yet been removed through cleaning. Thus, the impacts of using CADe, and its associated increase in FPs during the insertion phase, are unknown.

Some studies added those FPs that were detected by the CADe, considered to be a polyp, and removed by the endoscopist—but that were later determined to be normal tissue through pathological examination [14]. 

## 4. Studies That Report False Positives

### 4.1. Using CADe Based on Deep Learning for Real-Time Polyp Detection in Colonoscopy Videos

The details of these reports are summarized in Table 2. From the perspective of the clinical endoscopist, we only included studies evaluating CADe with the real-time capability to detect polyps. All CADe algorithms used in these studies were based on deep learning, which offered better sensitivity and specificity than the outdated hand-crafted-feature method [23,24]. Most of these studies were reporting on the development and validation of CADe systems with per-frame sensitivity between 56.8–98.8% and specificity between 63.3–98%.

In these reports, the false-positive rate (FPR) was reported either as a per-frame analysis [16,18] or a per-polyp analysis [15,17]. To determine the per-frame FPR, the videos were transformed into frames of still images. The per-frame FPR was typically calculated as the number of FP frames divided by the number of frames without polyps; However, a minority of the studies reported the per-frame FPR as the number of FP frames divided by the total number of frames [20,26]. Because of the minor difference in these two definitions, we categorized both as per-frame FPRs in the current review. When only the specificity of a CADe system was reported, the per-frame FPR was calculated as 1-per-frame specificity (i.e., the number of true negative frames divided by the number of frames without polyps). To calculate the per-polyp FPR, the number of polyps identified by the CADe but judged to not be polyps by the endoscopist was divided by the total number of polyps that were identified by the CADe. Reporting the per-polyp FPR is more realistic and clinically relevant. The FPR varied widely (per-frame, 0.9–37%). Most studies validated algorithms under the conditions of ideal bowel preparation for polyp detection. In a study with variable bowel preparation [15], the CADe model still performed well, with a per-polyp sensitivity of 98.8%; however, the per-polyp FPR was as high as 60%.

The FPR can be varied by adjusting its confidence level in the CADe algorithm. An image with a polyp detection probability greater than the confidence level is recognized by the CADe to be a polyp [16]. The CADe algorithm achieved a higher sensitivity of 94% for polyp detection and a higher per-frame FPR of 7.8% with a confidence level of ≥10%. When the confidence level was adjusted to ≥30%, the FPR decreased to 2.7% at the expense of reducing the sensitivity to 88%. Whenever possible, an ideal CADe system should strike a balance between high sensitivity and a low FPR [16].

### 4.2. RCTs Comparing Real-Time CADe with Control

The details of these reports are summarized in Table 3. All of the systems were based on deep learning convolutional neural networks. Out of these 6 studies, 5 were conducted in China, where the reported ADR was usually lower than those in the Western countries. All 6 studies consistently showed that the CADe increased the ADR [4,5,6,7,8,14], and most of the studies did not report missing any polyps [4,6,7,8]. All 6 studies saw an increase in the number of diminutive (<5 mm) adenomas, and some also saw an increase in the number of small (<10 mm) adenomas [4,5,6]. The withdrawal time, excluding biopsy, was comparable between the CADe and the control groups [4,5,7,8,14]. A false positive was defined in some reports as an area that was traced continuously but deemed not to be a polyp by the endoscopist [4,7,8] or was not defined in others [5,6,14]. False positives mainly consisted of bubbles, feces, and crumpled colon walls [4,7,14].

### 4.3. Video Analysis Studies Using FPs as the Primary Uutcome

The details of these reports are summarized in Table 4. False positives were identified as artifacts from the bowel wall and bowel content [12]. The clinical relevance of FPs was determined by the time required to explore the causes of FPs (in post hoc analysis). Most (94.3%) were bowel wall images with short exposure times and were determined to have no clinical impact, as no additional exploratory time was needed. 

Holzwanger et al. defined FPs according to time thresholds (0.5 s, 1 s, and 2 s) [13]. The different threshold definitions for FPs resulted in different reported diagnostic performances of the CADe, and the data suggested that using the same benchmarks to define FPs is the prerequisite for comparing the performances among different CADe algorithms.

## 5. The Causes of False Positives

The reported causes of false positives included feces, bubbles, wrinkled walls, normal structures (such as ileocecal valve), local inflammation, local bleeding, suction marks, polypectomy sites, and round drug capsules [4,7,15]. Of all these causes, the majority originated from rumpled colon folds, feces, debris, and bubbles (Table 1). The proportion of these elements depends on the CADe system used. Even with the same CADe algorithm, different settings (such as the confidence level) would result in different FPRs [16].

## 6. Adverse Effects of FPs

### 6.1. Increased Withdrawal Time

The time expended to differentiate an FP from a true lesion can potentially increase the withdrawal time. Although most RCTs on the real-time application of CADe found a longer withdrawal time in the CADe group compared to the control group [4,6,7,8], the withdrawal time without biopsy was not significantly different. Nonetheless, the withdrawal time without biopsy was numerically longer in all 6 RCTs (Table 3). In a post hoc analysis of a small fraction (40/342 or <11.7%) of the original CADe groups in the RCT studies, Hassan et al. found that 94% of FPs were discarded by the endoscopist immediately without further exploration, and the time wasted on the remaining FPs only contributed to about 1% of the withdrawal time. This was extrapolated to the original RCT data to suggest that FPs were insignificant, even though not all of the original FPs were assessed. Nevertheless, the mean withdrawal time was moderately correlated with the number of FP prompts (*p* = 0.0003; *r* = 0.5, 95% CI: 0.2–0.7) [12]. This positive correlation raised questions about the conclusion that FPs had no clinical impact [12]. Another study analyzing FPs also found that a higher number of FPs was associated with longer withdrawal time (Table 4) [13]. It appears that FPs did contribute to a longer withdrawal time, but that the impact might be quite limited by the commercially available system and experienced endoscopists. In a real-life situation, where the bowel preparation is usually less than optimal and endoscopists are less experienced, the impacts of bowel preparation on FPs and withdrawal time require more objective studies. 

### 6.2. Unnecessary Polypectomies of Non-Neoplastic Lesions

The presence of FPs might lead to unnecessary biopsies of non-neoplastic tissues. Of the 6 total, 4 RCTs [4,7,8,14] (with another 1 unreported [6] and 1 showing no difference [5]) listed in Table 3 showed a significant increase in the biopsy of non-neoplastic polyps in the CADe group, which was typically double the number reported for the control group. Hyperplastic polyps and inflammatory polyps were lumped together in these studies. The removal of hyperplastic polyps—other than the diminutive ones at the distal rectosigmoid colon—is justified, as these polyps contribute to the serrated pathway of colorectal carcinogenesis [27]. Therefore, it is unknown how many of the biopsies were really unnecessary. If these biopsies were, in fact, unwarranted, then there exists an avoidable non-indicated use of medical resources. Unnecessary biopsies could also add to the cost of pathology processing. 

The application of the CADx to characterize the polyps following their detection with the CADe might help reduce the number of unnecessary polypectomies of non-neoplastic polyps. Preliminary results showed promise for simultaneously classifying polyps with endocytoscopic images [28], or even with white light images [29] after using the CADe to detect the polyps in white light.

### 6.3. Increased User Fatigue, Distractions, and Decreased Enthusiasm

The recurrent appearance of FPs on the screen may lead to increased fatigue and decreased vigilance on the part of the endoscopist [30]. Vigilance is a limited resource and depletes with repetitive stimulus [31]. Hassan et al. reported the number of FPs far outnumbered that of true positives—a 25-fold difference [12]. Inundating the endoscopist with such a large amount of prompts on the screen, even if only very transient attention is demanded for each prompt, engenders the risk of the fatigue of the endoscopist. However, a study showed that a real-time CADe system, integrated on one primary endoscopy monitor instead of the two monitors used in most RCTs (Table 3), improved the ADR without an increase in the subjective fatigue level reported by the endoscopists during the colonoscopy [14]. The unblinded report, developed by proponents of the CADe algorithm under study, raised questions regarding the objectivity of the results.

False positives cause distractions and the need for refocusing, potentially resulting in adverse effects during the search for real polyps. To illustrate how difficult it is to refocus after distraction, a study on mobile phone use while driving showed that the risk of a rear-end accident occurring increased by 2.34–3.56 times, despite increasing their time headway by 0.41–0.59 s to offset the distraction of texting while driving [32].

Too many FPs may hamper the enthusiasm of the endoscopist to apply the CADe in clinical practice. One recent survey on the views of gastroenterologists regarding the potential use of artificial intelligence found that 33.9% of respondents worried about high numbers of FPs [33]. Reports that emphasize the lack of importance of FPs based on subjective assessment need to be re-evaluated by studies with more objective and unbiased designs.

## 7. How to Address the Occurrence of FPs

There is considerable variability in FPRs in the literature (Table 1). This variability suggests that there are diverse definitions of FPs and various conditions that affect the occurrence of FPs inside the bowel lumen, which indicates that there is an opportunity to minimize FPs through standardizing the definitions of FPs and optimizing the condition of the bowel lumen.

Standardizing the definitions of FPs will require agreement amongst programmers of the CADe system. An example of a simple method that could be used to reduce FPs is re-training the CADe algorithms with scenarios that currently lead to FPs. Another approach could be the adoption of recurrent neural networks, which have memory and can process temporal sequences of frames in a way that is similar to the learning process of human brains [10]. Misawa et al. reported that when they changed their old algorithm [17] to YoloV3 (You Only Look Once, Version 3), a state-of-the-art, real-time object detection algorithm, better specificity was achieved (increasing from 90.9% to 93.7%) [19]. Lee et al. proposed to reduce FPs by using the median filter (which reduced the FPR from 12.5% to 6.3%), a nonlinear spatial filter that is particularly effective for eliminating salt-and-pepper noise [21]. To filter out most short flashes, Podlasek et al. suggested setting a threshold of persistent time for FPs to show up; however, this method might introduce a minor detection lag, depending on the desired sensitivity [22]. These methods are beyond the expertise of the clinical endoscopists.

Optimal bowel preparation is the prerequisite for a high-quality CADe-assisted colonoscopy and is associated with fewer FPs [13]. As the major source of CADe FP alerts is the wrinkled walls, they can be reduced by ensuring adequate luminal insufflation. The use of an anti-spasmodic agent, such as Hyoscine-*n*-butylbromide, might be helpful in reducing the contraction of the colon wall [34]. Adding simethicone or rinse water to the bowel preparation regimen helps eliminate bubble-induced FPs [35,36]. However, whether the addition of these measures would actually decrease FPs remains to be studied.

Before the FPs can be effectively reduced, proper training of the endoscopist to recognize and ignore FPs is needed to enable the widespread adoption of the CADe for the detection of colon neoplasms [12].

The optimization of the condition of the bowel lumen can be controlled by the colonoscopist using water exchange colonoscopy, which will be discussed in detail below.

## 8. Water Exchange and Its Potential Beneficial Effect on Reducing FPs

Among the Gastrointestinal (GI) Endoscopy Editorial Board’s top 10 topics in endoscopy in 2019, water exchange (WE) and artificial intelligence (i.e., CADe) were both considered important advances in GI endoscopy [37]. The coincidence brought both to the forefront of the discussion on the improvement of ADR.

Compared with traditional gas (i.e., air or CO_2_) insufflation for colonoscopes, WE is an effective insertion method that minimizes insertion pain and enhances ADR [38,39,40]. It features infusing water to guide the scope advancement in an airless lumen, while suctioning the infused water at the same time during insertion, thus aiming at the almost complete removal of the infused water when cecal intubation is achieved. A network meta-analysis concluded that WE produced the highest ADR when compared with water immersion and gas insufflation [41]. A modified Delphi review also endorsed WE as having better bowel cleanliness, as well as less insertion pain and higher ADRs, than gas insufflation [42].

Holzwanger et al. reported that a high FPR was associated with fair or poor Aronchick bowel preparation scores [13]. WE can effectively salvage-clean bubbles and fecal debris during insertion, resulting in better bowel cleanliness during withdrawal. Table 5 summarizes 3 key RCTs, including more than 2000 patients in each group, comparing air insufflation and WE in terms of ADR. WE consistently showed better Boston Bowel Preparation Scale (BBPS) scores than air insufflation, both in the whole colon and the right colon, the latter of which was usually the dirtiest colon segment [38,39,40]. WE might also help reduce FPs associated with crumpled folds, as there is less need for suction cleaning, and thus the related spasms, during withdrawal [43]. In an analysis of the CADe-overlaid withdrawal phase videos of colonoscopies from an RCT comparing right colon ADR inserted with WE or air insufflation, Tang et al. found WE was associated with a significantly lower FPR compared with air insufflation (5 [4.1%] vs. 19 [15.6%], *p* = 0.02) (Dr. CP Tang, personal communication 2021).

Another potential merit of combining WE with CADe is the possible additive effects on increasing ADR. WE and CADe both increase ADR but through different mechanisms. WE increases ADRs mainly through insertion salvage cleaning, thus revealing otherwise unexposed polyps (Table 5). On the other hand, CADe works as a second observer and points out polyps that are exposed but not recognized due to human error [17]. A single-center study clearly demonstrated that even WE missed polyps in the right colon [44]. In other words, the individual strengths of WE and CADe complement the weakness of one another.

## 9. Conclusions

False positives have emerged as an important research issue in the application of artificial intelligence for the detection of polyps during colonoscopy. The number of FPs per colonoscopy turned out to be much higher (26 to 27 per colonoscopy) than originally reported among recent RCTs using the CADe to assist with polyp detection (0.07 to 0.2 per colonoscopy). This discrepancy might be a result of the different criteria for FPs that were set up in each study. A refinement of the definition of FPs is urgently needed to minimize variability in and facilitate the comparison of FPs reported in one study with those from another. A recurrent theme in published studies showed that a higher number of FPs was associated with less-than-ideal bowel preparation. The occurrence of FPs might result in the unnecessary biopsy of non-neoplastic lesions, which has been shown to be increased with the use of the CADe compared with the control groups in most RCTs. False positives might also lead to fatigue, distraction, and the need for refocusing for the endoscopists. Aside from re-training the CADe algorithms, adjunct medications (e.g., simethicone and an anti-spasmodic) might be beneficial for decreasing FPs. WE holds the potential for reducing FPs through salvage-cleaning feces and bubbles during insertion, thus avoiding cleaning and suction-induced colon spasming during withdrawal. The simultaneous application of the CADe and the CADx can help avoid unnecessary polypectomies of non-neoplastic polyps. Future studies on standardizing the definition and measurement of FPs are needed. Adding WE to CADe is a double-advantage approach, in that it may not only decrease FPs but may also further boost ADR to the benefit of the patients.

## Figures and Tables

**Figure 1 diagnostics-11-01113-f001:**
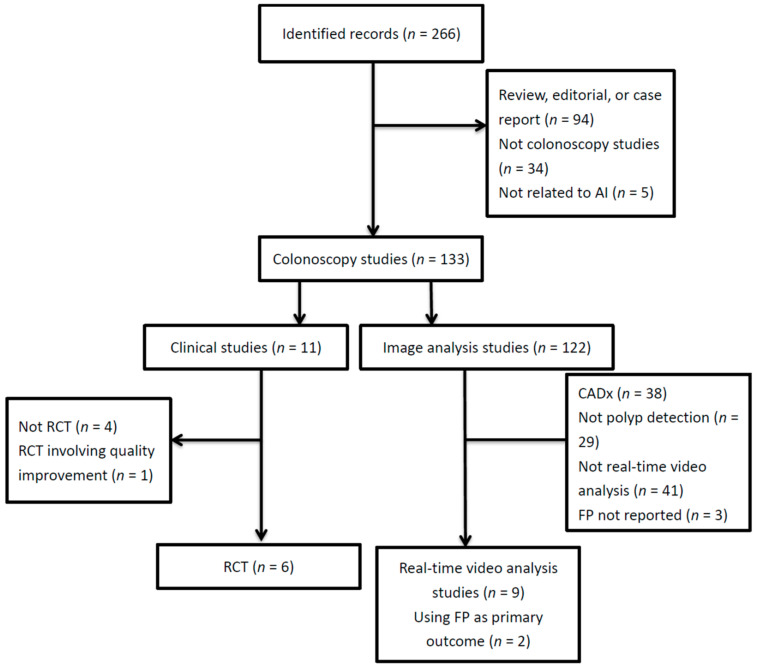
Literature flow diagram. AI, artificial intelligence; CADx, computer-aided diagnosis; FP, false positive; RCT, randomized controlled trial.

**Table 1 diagnostics-11-01113-t001:** Various definitions of false positives and false positive rates across studies.

Study	Per-Frame FPR	Per-polyp FPR	FPR per Colonoscopy	Causes of FP
Becq et al. [15]	NA	60%	NA	NA
Guo et al. [16]	When confidence ≥ 10%, 7.8% When confidence ≥ 30%, 2.8%	NA	NA	NA
Misawa et al. [17]	37%	60%	NA	NA
Urban et al. [18]	7%	NA	NA	NA
Misawa et al. [19]	6.3%	NA	NA	NA
Hassan et al. [20] *	0.9%	NA	NA	NA
Lee et al. [21]	8.3%	NA	19	NA
Podlasek et al. [22]	3%	NA	NA	NA
Wang et al. [7]	NA	NA	0.075	Feces and bubbles 66% Crumpled wall 18% Others 26%
Wang et al. [8]	NA	NA	0.1	NA
Su et al. [6]	NA	NA	0.201	NA
Liu et al. [4]	NA	NA	0.071	Feces and bubbles 64% Crumpled wall 19% Others 17%
Liu et al. [14]	NA	NA	0.074	Wrinkled mucosa 41% Feces 13.8% Bubbles: 10.3% Others: 34.5%
Holzwanger et al. [13]	NA	NA	26.3	Folds 91.8% Bubbles 5.6% Stool or others 2.5%
Hassan et al. [12]	NA	NA	27.3	Bowel wall 88% Bowel contents 12% (stools 5.8%, mucus 2.8%, bubble 2.3%, etc.)
Variability	41 folds	1 fold	338 folds	From bowel wall to feces and bubbles

* the number of false positive frames divided by the total number of frames; per-frame FPR, the number of false positive frames divided by the number of frames without polyps; per-polyp FPR, the number of polyps identified by the CADe but judged not to be polyps by the endoscopist is divided by the total number of polyps identified by the CADe. NA, not analyzed.

**Table 2 diagnostics-11-01113-t002:** Recent studies using CADe-overlaid videos for real-time detection of polyps.

Study	Primary Outcome	Videos Reviewed (*n*)	Polyps Detected	Sensitivity	Specificity
Misawa et al. [17]	Accuracy of CADe	155 positive videos and 391 negative videos. Most of the polyps were flat.	NA	Per-frame: 90%	Per-frame: 63.3%
Urban et al. [18]	Polyp detection by CADe	9 randomly selected colonoscopy videos	Performing endoscopist: 28 Three expert reviewers without CADe: 36 One expert reviewer with CADe: 45	Per-polyp: 94%	Per-frame: 93%
Becq et al. [15]	Polyp detection by CADe	50 colonoscopies from consecutive patients with various bowel preparations.	Performing endoscopist: 55 CADe: 401 possible polyps (100 definite polyps, 63 possible polyps, and 238 false positives	Per-polyp: 98.8%	NA
Guo et al. [16]	Accuracy of CADe	50 videos with small polyps and 50 videos without polyps.	NA	When confidence level ≥10%, per-frame: 66.9% When confidence level ≥30%, per-frame: 56.8%	When confidence level ≥10%, per-frame: 92% When confidence level ≥30%, per-frame: 98%
Wang et al. [25]	Accuracy of CADe	138 videos with polyps and 54 videos without polyps	NA	Per-frame: 91.6%	Per-frame: 95.4%
Misawa et al. [19]	Accuracy of CADe in a large, publicly accessible database.	100 videos	NA	Per-frame: 90.5% Per-polyp: 98.0%	Per frame: 93.7%
Hassan et al. [20]	Accuracy of CADe	138 polyp-positive short videos	NA	Per-frame: 99.7%	NA
Lee et al. [21]	Accuracy of CADe	15 unaltered videos	Performing endoscopist: 38 CADe: 45	Per-frame: 89.3%	NA
Podlasek et al. [22]	Accuracy of CADe	42 colonoscopy videos	Reviewer: 84 CADe: 79	Per-polyp: 94.1%	NA

CADe: computer-aided detection; NA, not analyzed.

**Table 3 diagnostics-11-01113-t003:** Recent RCTs comparing real-time CADe with control on adenoma detection during colonoscopy.

Study	Location of Study	Control vs. CADe (*n*)	Overall ADR	Non-Neoplastic Polyps Detected, *n* (%)	CADe Used During Insertion	Number of Screens Used	Withdrawal Time, Mean, Minutes	Withdrawal Time, Exclude Biopsy, Mean, Minutes
Wang et al. [7]	China	536 vs. 522	20.3% vs. 29.1% *	94 (34.9) vs. 217 (43.6) * (hyperplastic plus inflammatory)	No	2	6.39 vs. 6.89 *	6.07 vs. 6.18
Wang et al. [8]	China	478 vs. 484	28% vs. 34% *	113 (37) vs. 200 (40) * (hyperplastic plus inflammatory)	No	1	6.99 vs. 7.46 *	6.37 vs. 6.48
Repici et al. [5]	Italy	344 vs. 341	40.4% vs. 54.8% *	57 (16.6) vs. 68 (19.9) (Normal, hyperplastic, inflammatory and others)	Yes	1	NA	7.0 vs. 7.3
Su et al. [6]	China	308 vs. 315	16.5% vs. 28.9% *	NA	No	2	5.68 vs. 7.03 *	6.74 vs. 6.82 *
Liu et al. [4]	China	518 vs. 508	23.9% vs. 39.1% *	92 (37.1) vs. 203 (41.8) * (proliferative and inflammatory)	No	2	NA	6.32 vs. 6.37
Liu et al. [14]	China	397 vs. 393	20.9% vs. 29.0% *	87 (42.7) vs. 222 (52.7) * (hyperplastic and inflammatory)	No	1	6.94 vs. 7.29 *	6.62 vs. 6.71

ADR, adenoma detection rate; CADe, computer-aided detection; FP, false positive; FPR, false positive rate; NA, not analyzed; RCT, randomized controlled trial; *, statistically significant; NA, not analyzed.

**Table 4 diagnostics-11-01113-t004:** Video analysis studies using FPs as the primary outcome.

Relevant Data	Holzwanger et al. [13]	Hassan et al. [12]
Manufacturer of CADe model	Shanghai Wision AI Co., Ltd.	GI-GENIUS, Medtronic, Version 1.0.2. June 2019
Primary outcome	FPs per colonoscopy	To generate a structured classification of FPs and to estimate their frequency and clinical relevance
Videos reviewed (*n*)	62 colonoscopy videos collected prospectively with consecutive patients undergoing routine colonoscopy	A post hoc analysis of 40 withdrawal phase videos of the CADe arm from an RCT

CADe, computer-aided detection; FP, false positive; RCT, randomized control trial; SD, standard deviation.

**Table 5 diagnostics-11-01113-t005:** BBPS scores in key randomized controlled trials comparing ADRs between WE and air insufflation.

Study	Sample Size, Air Insufflation vs. WE (*n*)	Primary Outcome: ADR (95%CI)	Overall BBPS Scores or	Right Colon BBPS Score
Jia et al. [40]	1650 vs. 1653	13.4% vs. 18.3%; RR 1.45 (1.20–1.75) *	7.0 ± 2.3 vs. 7.3 ± 1.6 ^#^ (Mean ± SD)	2.3 ± 0.7 vs. 2.2 ± 1.5 ^#^
Hsieh et al. [39]	217 vs. 217	37.5% (31.6–44.4%) vs. 49.8% (43–56.4%) *	6.2 ± 1.1 vs. 7.1 ± 1.3 ^#^ (Mean ± SD)	NA
Cadoni et al. [38]	408 vs. 408	43.4% (35.6–45.3 %) vs. 49.3% (44.3 –54.2 %) *	8.0 (6.0–9.0) vs. 9.0 (7.0–9.0) ^#^ [Median (IQR)]	2.0 (2.0–3.0) vs. 3.0 (2.0–3.0) ^#^

ADR, adenoma detection rate; BBPS, Boston Bowel Preparation Scale; CI, confidence interval; IQR, inter-quartile range; RR, relative risk; SD, standard deviation; WE: water exchange; *, *p* < 0.05; ^#^, *p* < 0.001; NA, not analyzed.

## Data Availability

All literatures reviewed are accessible on PubMed.

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
