# Peer review of "Computer-Aided Detection False Positives in Colonoscopy"

_diagnostics, 2021, doi:10.3390/diagnostics11061113_

Round 1
Reviewer 1 Report
I think your paper deals with an important and modern topic. The paper is well performed, but it is a little confusing because of many subchapters that could be too long for the readers. So I think the paper should be shortened. Even in the title I suggest to eliminate the problem of water exchange focusing only on the problems of false positives.
The first sentences of introduction should be changed.
Author Response
Point-to-point responses to the comments
Reviewer 1:
I think your paper deals with an important and modern topic.
Response: Thank you for your positive comment.
The paper is well performed, but it is a little confusing because of many subchapters that could be too long for the readers. So I think the paper should be shortened.
Response: We have shorted the manuscript. Please see the tracked changes.
Even in the title I suggest to eliminate the problem of water exchange focusing only on the problems of false positives.
Response: We have eliminated “water exchange” from the title as the following: “Computer-aided detection false positives in colonoscopy.”
The first sentences of introduction should be changed.
Response: WE have changed the first paragraph of introduction as the following, “Missed lesions account for 57.8% of interval colorectal cancers – cancers that occur within 3-5 years after a negative colonoscopy [1]. To reduce missed lesions and the incidence of interval cancer, measures were proposed to improve quality of colonoscopy. One of the most important quality metrics is adenoma detection rate (ADR), defined as proportion of patients with at least one adenoma [2].”
Reviewer 2 Report
Thank you for your journal submission. Systematic reviews and meta-analyses can be especially time consuming and can be a challenge. Your time and effort for researching this topic is greatly appreciated. Please find my comments below with corresponding line numbers.
52: possible typo: systemic instead of systematic?
52-63: the methods section is lacking with regards to the search process. i have never seen an example where a study found xxx many papers and all xxx was utilized. my guess is that papers not meeting your criteria was found. was the PRISMA guideline referred to? http://www.prisma-statement.org/ Please refer to the checklist and flow diagram for new systematic reviews that only search databases.
the purpose of this process is to be as transparent as possible and to increase the reproducibility of your process.
table 5 (approx 328 - 329):
- Air Insufflation vs WE column, sample size (n) is missing
- ADR column, please include confidence intervals from the papers if they were reported
- please indicate which is mean and which is median
347: to my knowledge there is no widely accepted definition of poor/fair bowel prep. I have seen some variation in these definitions, especially with regard to BBPS. Could you please elaborate. If this was already done in the paper, I apologize. Could you please direct me to the appropriate line?
357: The word 'unique' is often misused. Are you sure that this combination of technologies is literally the only way to jointly reduce FPs and increase ADR?
Author Response
Point-to-point responses to the comments
Review 2:
Thank you for your journal submission. Systematic reviews and meta-analyses can be especially time consuming and can be a challenge. Your time and effort for researching this topic is greatly appreciated. Please find my comments below with corresponding line numbers.
Response: Thank you for your kind comments.
52: possible typo: systemic instead of systematic?
Response: We’ve corrected to “systematic review”
52-63: the methods section is lacking with regards to the search process. i have never seen an example where a study found xxx many papers and all xxx was utilized. my guess is that papers not meeting your criteria was found. was the PRISMA guideline referred to? http://www.prisma-statement.org/ Please refer to the checklist and flow diagram for new systematic reviews that only search databases.
the purpose of this process is to be as transparent as possible and to increase the reproducibility of your process.
Response: We re-write the method section and add a literature flow diagram, “We performed a systematic review of the literature by searching PubMed with the following string: (automatic polyp detection OR computer aided detection OR deep learning OR artificial intelligence) AND colonoscopy in the past four years (Jan. 2017 to Apr. 2021). Search was performed on title, abstract and keywords. We included full text articles in English. The exclusion criteria were non-research reports (systematic review, editorial, or case report), research not related to artificial intelligence, not focusing on colonoscopy (computed tomography colonography, capsule endoscopy, chatbot, etc.), not for polyp detection (CADx, regulatory issues, robotic colonoscopy, quality optimization, etc.), not applying real-time video analysis, or not reporting FP. For clinical studies, non-RCT was excluded. We identified 9 articles on application CADe based on deep learning for real-time detection of polyps on colonoscopy videos, 6 articles on RCTs comparing colonoscopies with or without CADe to assist polyp detection, as well as 2 CADe overlaid video analysis studies using FP as the primary outcome (Figure 1). We turn to one of our most recent reviews on WE and tabulated 3 articles on RCTs comparing WE with air insufflation, especially FP- related procedural data [11]. ”
table 5 (approx 328 - 329):
Air Insufflation vs WE column, sample size (n) is missing
Response: the column head has been changed to “Sample size, air insufflation vs. WE (n)”
ADR column, please include confidence intervals from the papers if they were reported
Response: Confidence intervals have been added to cells of the ADR column.
please indicate which is mean and which is median
Response: the “mean” and “median” have been added to their proper places.
The revised Table 5:
|
|
Sample size, air insufflation vs. WE (n) |
Primary outcome: ADR (95% CI) |
Over all BBPS scores |
Right colon BBPS score |
|
Jia et al. [40] |
1650 vs. 1653 |
13.4% vs. 18.3%; RR 1.45 (1.20–1.75)* |
7.0±2.3 vs. 7.3±1.6 (mean ± SD)# |
2.3±0.7 vs. 2.2±1.5 # |
|
Hsieh et al.[39] |
217 vs. 217 |
37.5% (31.6%-44.4%) vs. 49.8% (43.2%-56.4%)* |
6.2±1.1 vs. 7.1±1.3 (mean ± SD)# |
NA |
|
Cadoni et al.[38] |
408 vs. 408 |
43.4% (35.6 % - 45.3 %) vs. 49.3% (44.3 % - 54.2 %)* |
8.0 (6.0 – 9.0) vs. 9.0 (7.0 – 9.0) [median (IQR)]# |
2.0 (2.0 – 3.0) vs. 3.0 (2.0 – 3.0) # |
347: to my knowledge there is no widely accepted definition of poor/fair bowel prep. I have seen some variation in these definitions, especially with regard to BBPS. Could you please elaborate. If this was already done in the paper, I apologize. Could you please direct me to the appropriate line?
Response: We clarify the definition as the following, “a fair or poor Aronchick bowel preparation scores”
357: The word 'unique' is often misused. Are you sure that this combination of technologies is literally the only way to jointly reduce FPs and increase ADR?
Response: We agree with you and modified the sentence in question to “Adding WE to CADe is a double-advantage approach in that it may not only decrease FP but also further boosts ADR to the benefit of the patients”

Round 2
Reviewer 2 Report
Thank you for your clear and concise responses to my comments.